# Automatic Segmentation of Standing Trees from Forest Images Based on Deep Learning

**DOI:** 10.3390/s22176663

**Published:** 2022-09-03

**Authors:** Lijuan Shi, Guoying Wang, Lufeng Mo, Xiaomei Yi, Xiaoping Wu, Peng Wu

**Affiliations:** 1College of Mathematics and Computer Science, Zhejiang A&F University, Hangzhou 311300, China; 2School of Information Engineering, Huzhou University, Huzhou 313000, China

**Keywords:** semantic segmentation, deep learning, attention mechanism, standing tree image

## Abstract

Semantic segmentation of standing trees is important to obtain factors of standing trees from images automatically and effectively. Aiming at the accurate segmentation of multiple standing trees in complex backgrounds, some traditional methods have shortcomings such as low segmentation accuracy and manual intervention. To achieve accurate segmentation of standing tree images effectively, SEMD, a lightweight network segmentation model based on deep learning, is proposed in this article. DeepLabV3+ is chosen as the base framework to perform multi-scale fusion of the convolutional features of the standing trees in images, so as to reduce the loss of image edge details during the standing tree segmentation and reduce the loss of feature information. MobileNet, a lightweight network, is integrated into the backbone network to reduce the computational complexity. Furthermore, SENet, an attention mechanism, is added to obtain the feature information efficiently and suppress the generation of useless feature information. The extensive experimental results show that using the SEMD model the MIoU of the semantic segmentation of standing tree images of different varieties and categories under simple and complex backgrounds reaches 91.78% and 86.90%, respectively. The lightweight network segmentation model SEMD based on deep learning proposed in this paper can solve the problem of multiple standing trees segmentation with high accuracy.

## 1. Introduction

Semantic segmentation of standing trees is crucial to the obtaining of factors such as tree crown and diameter at breast height (DBH) from standing tree images, which is an important part of related researches in the field of “digital forestry” [1]. Martins et al. proposed a region-based convolutional neural network (CNN) object instance segmentation algorithm for semantic segmentation of tree canopies in urban environments [2]. Airborne high-resolution urban forest images are mainly used for segmentation, which is costly. Yao et al. used fuzzy logic system (FLS) for semantic segmentation of standing trees in satellite images [3]. Chen et al. used the adversarial network Pix2Pix to segment partially occluded apple tree images, which was based on the depth images of apple trees captured by an RGB-D camera [4]. Ge et al. used a hybrid active contour model driven by pre-fitting energy to adaptive sign function for fast image segmentation [5,6].

With the continuous maturity of deep learning, it has been widely used in image processing, target detection and other fields in recent years to improve the reasoning ability of image models and strengthen the learning ability of the network. In the field of image segmentation, a variety of segmentation models have been proposed, which effectively solve a series of problems in traditional segmentation, such as semi-manual operation [7], inaccurate segmentation [8], and inaccurate target subjects [9].

The process of standing tree image segmentation is shown in Figure 1, which mainly includes two stages: training and application. In the training stage, the standing tree images are captured by the cameras firstly, and the data augmentation method is used to expand them. Then, the images are labelled to construct the standing tree image training dataset. After this, the training dataset is inputted to train the semantic segmentation network. When the loss value approaches convergence, training stops, and a trained segmentation model is obtained. In the application stage, the target standing tree image is input into the trained model, and the segmentation result is obtained.

In order to solve the problem of inaccurate segmentation of multiple standing trees in complex backgrounds, based on the powerful detail processing function of the deep learning neural network DeepLabV3+, combined with the unfavorable factors, such as the diverse forms of standing trees and the complex growth environment, a lightweight network segmentation model based on deep learning, SEMD (SENet and MobileNet embedded in DeepLabV3+), is proposed in this paper. The model is based on the DeepLabV3+ [10] framework, which is used to collect multi-scale information of standing tree images and effectively reduce the loss of feature information. In SEMD, the backbone network is replaced by a lightweight convolutional neural network MobileNet which effectively reduces computational complexity and hardware requirements. Furthermore, the SENet [11] attention mechanism is added into SEMD to improve the effective reading of feature point information and suppress the generation of useless feature information, so as to focus on the main body of the standing tree and avoid inaccurate segmentation due to low image pixel resolution.

The remaining parts of this paper are organized as follows. Section 2 describes the main idea of the SEMD model proposed in this paper and the basic principles involved in detail. Section 3 describes the experimental setup and experimental procedure. Section 4 analyzes the comparative experimental results. Finally, Section 5 carries on with the conclusions of the work.

## 2. SEMD Model

### 2.1. Main Idea

SEMD, a lightweight network segmentation model based on deep learning, proposed in this paper is shown in Figure 2.

According to the characteristics of accurate segmentation of multi-limbs with complex backgrounds, the main ideas of SEMD are as following.

(1)Being based on DeepLabV3+ framework. Using the unique spatial pyramid structure of DeepLabV3+ network structure, the convolutional features of the standing tree image can be multi-scale fused, so as to reduce the loss of edge details of the standing trees. The DeepLabV3+ semantic segmentation model mainly includes two parts: encoder and decoder. The encoder consists of a backbone network Xception, a spatial pooling module atrous spatial pyramid pooling (ASPP), and an up-sampling module. The decoder processes the data transmitted by the encoder at different down-sampling layers to output feature information.(2)Replacing backbone network with MobileNet. Compared to Xception, the backbone network originally used by DeepLabV3+, MobileNet is a lightweight convolutional neural network, which can effectively reduce the computational complexity and hardware equipment requirements and improve the computational speed of the model.(3)Importing attention mechanism SENet. The attention mechanism SENet can improve the effective reading of feature point information and suppress the generation of useless feature information, so as to focus on the main body of the standing tree and avoid inaccurate segmentation due to low image pixel resolution.

In SEMD, MobileNet+SENet is used in the encoder part as the backbone network. Using this structure, features can be extracted and calculated with any resolution. The ratio of the spatial resolution of the input image to the spatial resolution of the output image is used as the output stride. Multi-scale fusion of the features calculated by MobileNet+SENet using different atrous convolutions effectively reduces the executing time without affecting the segmentation accuracy. The decoder part mainly uses the same convolutional network to reduce the number of information channels and solve the problem of difficulty in training due to too many low-feature channels.

### 2.2. Being Based on DeepLabV3+ Framework

Compared with the previous DeepLab series networks, the advantage of DeepLabV3+ is that it introduces an encoder that can control the extraction of arbitrary feature resolution, and balances accuracy and time-consuming through hole convolution. To solve the problem that the signal is continuously down-sampled and the details are lost during the encoding process, DeepLabV1 [12] uses fully connected conditional random field (CRF) to improve the ability of the model to capture structural information and solve the problem of fine segmentation, but there are still some limitations. DeepLabV2 [13] replaces the backbone network VGG16 used by V1 with Resnet101 and proposes the concept of atrous spatial pyramid pooling (ASPP). ASPP can be used to address the difference in size of different detection targets [14]. However, since ASPP uses a 3 × 3-hole convolution with a large sampling rate, the image boundary cannot capture long-distance information, resulting in a degenerate into a 1 × 1 convolution. To overcome this problem, DeepLabV3 proposes a more general framework, adding a batch normalization (BN) layer, and improving ASPP to replace the original hole convolution of size 3 × 3, dilation = 24 with an ordinary 1 × 1 convolution, with the effective weights in the middle part of the filter are retained, and global average pooling is added to better capture global information. Without increasing the amount of computation, the range of information contained in each output convolution is enlarged, the receptive field is increased, and the speed and accuracy of semantic segmentation are improved, as is shown in Figure 3. DeepLabV3+ summarizes some of the above advantages, and the backbone network is highly adaptable. Therefore, for better segmentation results, some researchers replaced the backbone network with a residual network (ResNet) and a depthwise separable convolution (Xception), which leads to model complexity. Continuous improvement, which means that the requirements for mobile terminals or embedded devices are relatively high, which cannot meet the application of multiple scenarios.

Yang et al. used DeepLabV3+ to pedestrian segmentation and realized multi-scene applications [15]. Fu et al. proposed the semantic segmentation of bridge cracks based on DeepLabV3+ to achieve more accurate segmentation of crack details, which indicated that DeepLabV3+ has better edge segmentation capabilities [16]. Peng et al. proposed the semantic segmentation of litchi branches based on the DeepLabV3+ model, and used the Xception feature extraction model as the backbone network to achieve smaller branch segmentation, but the backbone network is complex and requires high hardware processing capabilities [17].

The model in this paper, SEMD, selects DeepLabV3+ network with strong adaptability of the underlying network, and its structure is mainly divided into an encoding layer and a decoding layer. The encoding layer is used for feature extraction and loss reduction, and the decoding layer is used to extract details and recover spatial information.

### 2.3. Replacing Backbone Network with MobileNet

The original DeepLabV3+ backbone network uses Xception as the backbone network. Due to continuous downsampling and pooling operations, the feature dimension of the network will gradually increase, and with a large amount of invalid feature dimension information, the accuracy of segmentation will be reduced. Focusing on this problem, a lightweight neural network, MobileNet, is adopted in SEMD model of this paper as the substitution backbone network.

There are three versions of MobileNet: MobileNetV1 [18], MobileNetV2 [19], and MobileNetV3 [20]. In this paper, MobileNetV2 is adopted for a moderate performance. MobileNetV2 is a lightweight feature extraction network that mainly consists of depthwise separable convolutional kernels (bottleneck) and an inverted residuals structure with linear bottleneck.

(1) Depthwise separable convolutions: It can effectively reduce the number of parameters and computation of the model, and the process is shown in Figure 4.

The input feature map, *F_H_*, *F_W_* are the height and width of the feature map respectively, *M* is the number of channels, and the convolution operation is performed on the feature map with a convolution kernel size of *K × K* and a number of convolution kernels of *N*. In the process of standard convolution of the feature map, the computational quantity *C*_1_ and the parametric quantity *P*_1_ are defined in Equations (1) and (2).
(1)$$C_{1}=K\times K\times M\times N\times F_{H}\times F_{W},$$
(2)$$P_{1}=K\times K\times M\times N+K\times K\times N.$$

The depthwise separable convolutions operation is performed on the feature map, and the computational quantity *C*_2_ and the parametric quantity *P*_2_ are defined in Equations (3) and (4).
(3)$$C_{2}=K\times K\times M\times F_{H}\times F_{W}+M\times N\times F_{H}\times F_{W},$$
(4)$$P_{2}=K\times K\times M\times N+K\times K\times N.$$

The ratio of the computational effort of depthwise separable convolutions able to standard convolutions and is defined in Equation (5).
(5)$$\frac{C_{2}}{C_{1}}=\frac{1}{N}+\frac{1}{K^{2}}.$$

(2) Inverted residuals. In the traditional residual structure block, the tensor depth containing the input feature map is first subjected to 1 × 1 convolution to reduce the dimension, and then the output is provided to the 3 × 3 convolution for dimensional increase operation. However, the features extracted in the depthwise convolution layer are limited to the input feature dimension. If the residual structure is used, the 1 × 1 point-by-point convolution dimension reduction operation is performed first, and the input feature map is compressed, and then the depthwise convolution extraction is performed, which leads to less extracted effective features. Therefore, MobileNetV2 first performs a 1 × 1 point convolution and dimension-raising operation to expand the channels of the feature map, enrich the number of features, and improve the accuracy of feature extraction. It is proposed on the one hand to ensure the accuracy of the results, and on the other hand to solve the problems in the process of image training due to too large neural network, complex calculation, insufficient hardware equipment, etc., so as to better adapt to the mobile terminal.

Inverted residuals and a new linear activation function ReLU6 [21] are adopted on the basis of MobileNetV1 network. ReLU6 means that the output value remains unchanged when the output value is between 0 and 6, and it is uniformly judged as 6 when the output exceeds 6, which solves the problem of serious information loss when the number of channels is small. It is defined in Equation (6).
(6)ReLU6=min(max(0,x),6).

Since the depth convolution cannot change the number of channels, it can only work in low dimensions, so the effect is not good. According to this, the point convolution is used to increase the dimension (the multiple is 6), and the convolution operation is performed in a high dimension to extract features, and finally through the point convolution restores the channel count reduction dimension. The network structure of MobileNetV2 is shown in Table 1. The operations include normal convolution (conv2d), depthwise separable convolutions (bottleneck), and average pooling (Avgpool). In the table, *t* represents the expansion factor, *c* represents the number of output channels, *n* represents the number of repetitions of the convolutional layer, *s* represents the stride, and spatial convolution uses 3 × 3 convolution kernel.

### 2.4. Importing Attention Mechanism SENet

The SENet module is designed to enable the network to perform dynamic channel feature recalibration to improve the representational capability of the network, whose structure is shown in Figure 5. Simply put, it is to automatically obtain the importance of each feature channel through learning, and then use this result to improve useful features and suppress features that are not useful for the current task. It is mainly divided into three parts [21]: 

Squeeze operation: After obtaining *U* (multiple feature maps), each feature map is compressed by the global average pool, so that the *C* feature maps become a 1 × 1 × *C* real number sequence.

Excitation operation: Use a fully connected neural network to perform a nonlinear transformation on the result after Squeeze. 

Reweight operation: Use the result obtained by Excitation as a weight and multiply it to the input feature. Its mapping relationship is shown in Equations (7)–(9).
(7)$$z_{c}=F_{sq}\left(u_{c}\right)=\frac{1}{H\times W}\sum_{i=1}^{H}\sum_{j=1}^{W}u_{c}\left(i,j\right),$$
(8)$$s=F_{ex}\left(z,W\right)=\sigma \left(g\left(z,W\right)\right)=\sigma \left(W_{2}\delta \left(W_{1}z\right)\right)\frac{1}{n},$$
(9)$$\tilde{x_{c}}=F_{scale}\left(u_{c},s_{c}\right)=s_{c}u_{c}.$$

Among them,$\;u_{c}$ represents each feature channel; *W* and *H* represent the width and height of$\;u_{c}$, respectively; $z_{c}$ represents the compression value of the *c*-th dimension channel; $W_{1}\;$and $W_{2}$, which mean the dimension decrease and increase respectively, are the weights of the full connection operation; δ is the ReLU function and σ is the sigmoid function. The function δ represents the first fully connected layer and σ is the second fully connected layer. Finally, by multiplying the scalar $s_{c}$ with the feature$\;\;u_{c}$, $\tilde{x_{c}}$, which represents a *c*-dimensional final feature, is obtained.

MobileNet is a lightweight depth wise separable convolutional neural network, which can greatly reduce the amount of parameters and computational complexity when dealing with classification problems of large and small features, while ensuring that the loss of accuracy is reduced [22]. In addition, the SENet attention mechanism module can also make good use of the relationship between channels to perform flexible and accurate calibration of channel features. Inspired by this, SEMD model proposed in this paper combines SENet module with MobileNet and imports them into the DeepLabV3+ architecture for pretraining. The traditional MobileNet network is optimized to improve the ability to learn subtle features in standing tree images under different backgrounds.

Using SENet as part of the search space leads to a more stable architecture. However, because the training will consume more time after adding the SENet structure, the channel of the extension layer is changed to one quarter of the original with little effect on the accuracy. This not only improves accuracy, but also reduces training time. The optimized network structure is shown in Table 2.

After removing the last original layer of MobileNet, the SENet module is added behind MobileNet with an additional 3 × 3 × 512 convolutional Layer for high-dimensional feature extraction. The fully connected layer is replaced with a global average pooling layer and a fully connected layer with ReLU activation function is used to enhance the feature extraction ability of stumbling edge details. The model is trained twice for transfer learning. In the first step, the bottom convolutional layer is trained using pre-trained weights on ImageNet. In the second step, the required weights are obtained by loading the pre-trained model in the first step and adjusting the parameters using the standing tree data set. Use self-trained weights for stump prediction segmentation.

## 3. Experiments

In this section, the results of the SEMD model proposed are experimented and verified. The hardware and software configurations, network hyper-parameter settings, stranding tree image dataset required for this experiment, and experimental procedure are introduced.

### 3.1. Hardware and Software Configuration 

The deep learning framework PyTorch is used to implement the SEMD model and the experiments. The hardware and software configurations are shown in Table 3. 

### 3.2. Dataset

#### 3.2.1. Image Data Acquisition

In different weather conditions, a dataset with 778 standing tree images was collected using the camera of a mobile phone, HONOR 30. The dataset includes 179 images of a single standing tree in a simple background, 179 images of a single standing tree in a relatively complex background, 210 images of multiple standing trees in a simple background, and 210 images of multiple standing trees in a relatively complex background. The varieties of standing trees include sweet-scented osmanthus, ginkgo, camphor, and so on. The sizes of images are all 2736 × 2736 pixels.

#### 3.2.2. Image Data Preprocessing

The contents of the collected data samples vary greatly, which is convenient to enhance the robustness of the detection network.

In order to compare the performance of SEMD with that of other related segmentation methods under the same condition and improve the efficiency of segmentation experiments, the images are unified, and the image size is adjusted to 512 × 512 pixels in the following experiments except Section 4.4.

On the other hand, in order to verify the robustness of the SEMD model, the image is additionally adjusted to 321 × 321 pixels and 415 × 415 pixels.

#### 3.2.3. Image Data Augmentation

Data augmentation plays an important role in semantic segmentation. Effective data augmentation can improve the robustness of the model and obtain higher generalization ability. In order to improve the segmentation accuracy and enhance data training, methods, such as random rotation and proportional flip, are used to obtain twice the amount of data, which is shown in Figure 6.

#### 3.2.4. Data Annotation

The images in the dataset were labelled using the open-source tool LabelMe [23]. LabelMe is a graphical interface image labeling software that can label polygons, rectangles, circles, polylines, line segments and points. The label format is JSON, which is converted into a single-channel image in subsequent processing and stored in the PASCAL VOC [24] data format.

### 3.3. Experiments Design

#### 3.3.1. Determination of Training Set and Test Set

Based on the above dataset construction, the standing tree images in simple and complex backgrounds are randomly allocated by function, 90% are training sets, and 10% are test sets. Due to the existence of a certain randomness, multiple tests are required. Both simple and complex image backgrounds in the training set include single and multiple standing trees to improve the accuracy.

#### 3.3.2. Evaluation Indicators

In the segmentation model, MIoU (Mean Intersection over Union) and MPA (Mean Pixel Accuracy) are used as important evaluation indicators for segmenting standing trees. The definitions of MIoU and MPA are shown in Equations (10) and (11).
(10)$$MIoU=\frac{1}{k+1}\sum_{i=0}^{k}\frac{p_{ii}}{\sum_{j=0}^{k}p_{ij}+\sum_{j=0}^{k}p_{ji}-p_{ii}},$$
(11)$$MPA=\frac{1}{k+1}\sum_{i=0}^{k}\frac{p_{ii}}{\sum_{j=0}^{k}p_{ij}}.$$

In the formulas, within a total of *K + 1* classes ($L_{0}\;$to $L_{k}$, which contains an empty class or background), *i* means the ground truth, *j* means predicted segmentation, and $p_{ij}$ means predict class *i* as the number of pixels in class *j*.

In order to test whether the standing tree image segmented by this model has the seriousness of distortion problem, this paper adopts PSNR (Peak Signal to Noise Ratio) [25] as the objective evaluation index, which is currently the most widely used image evaluation metric. Its unit is *dB*. The larger the value is, the weaker the image distortion is, the better the image segmentation performance is, and the higher the usability of the segmentation method is. PSNR is simply calculated by means of mean square error (MSE). The formula of mean square error is Equation (12), and the definition of peak signal-to-noise ratio is shown in Equation (13), where *H* and *W* are the height and width of the image, respectively; *n* is the number of pixel bits.
(12)$$MSE=\frac{1}{H\times W}{\sum_{i=1}^{H}\sum_{j=1}^{W}\left(X\left(i,j\right)-Y\left(i,j\right)\right)}^{2},$$
(13)$$PSNR=10\log_{10}\left(\frac{{\left(2^{n}-1\right)}^{2}}{MSE}\right).$$

#### 3.3.3. Experimental Scheme

In order to test the segmentation effect of the SEMD model on standing trees in both simple and complex backgrounds, the same experimental parameters are used in this paper.

(1)Optimization algorithm

There are many kinds of training algorithms in deep learning networks. This paper selects the Adam optimization algorithm [26], which is computationally efficient and can naturally realize learning rate adjustment.

(2)Loss function

The loss function is a combination of ordinary cross entropy loss (Cross Entropy Loss) and Dice Loss. The formula definition is shown in Equation (14). $\;p_{n,c}\in P\;$and are $y_{n,c}\in Y\;$the target label and prediction probability of class C and nth pixel in batch processing respectively. *Y* and *P* represent image truth and prediction results. *C* and *N* are the number of classes and pixels of the dataset in batch processing. The common cross-entropy function measures the difference of probability segmentation. There is a softmax function to classify pixels and use it, and the Dice coefficient is a set of similarity measurement functions used to calculate the similarity of two samples. The value is generally between 0 and 1. The lower the loss value, the better the fitting effect of the segmentation model, and the better the robustness.
(14)$$\varepsilon \left(Y,P\right)=-\frac{1}{N}\sum_{c=1}^{C}\sum_{n=1}^{N}\left(y_{n,c}{\log p}_{n,c}+\frac{{2y}_{n,c}p_{n,c}}{y_{n,c}^{2}p_{n,c}^{2}}\right).$$

(3)Training parameters

During training, the same set of training parameters are used, all defined according to the analysis above. Setting the initial learning rate Lr = 5 × 10^−5^, after 100 times of training, the model loss value is as low as 0.13, and the loss value of the loss function is shown in Figure 7. Therefore, the maximum number of training iterations is set to epoch = 100, 100 steps per train, and 50 steps for val. The batch size of this model is set to 8 due to the limitation of GPU memory. Some of the training hyper-parameters of the SEMD model are shown in Table 4.

Select standing tree images and use the above parameters for training. During the training process, the loss rate of the saved results and the model are used to predict the validation data of 512 × 512 pixels and overlay it with the original image. The results are shown in Figure 8.

The input size is an appropriate parameter selected from the comprehensive consideration of the SEMD model and the complexity of the standing tree image.

## 4. Results and Discussion

### 4.1. Visual Analysis

Based on our standing tree dataset introduced above, which includes two cases: simple background standing tree images and complex background standing tree images for verification, experiments are performed using SEMD model which is proposed in this paper, and some other methods such as FCN [27], SegNet [28], U-Net [29], PSPNet [30], and the original DeepLabV3+. The comparisons of segmentation results are shown in Figure 9 and Figure 10. Among them, the first column of Figure 9 and Figure 10 are the original images of standing trees, the second column is the ground-truth, the third column is the FCN model segmentation result, the fourth column is the SegNet Model segmentation results, the fifith column is the U-Net model segmentation result, the sixth column is the PSPNet model segmentation result, the seventh column is the original DeepLabV3+ model segmentation result, and the eighth column is the segmentation result of SEMD model proposed in this paper.

#### 4.1.1. Experiment on Standing Trees with Simple Background

The results of standing tree segmentation under simple backgrounds using different methods are shown in Figure 9. 

From the different network models for the segmentation of standing trees in simple background, FCN segmentation is relatively poor and can only roughly segment the shape of the canopy. SegNet network cannot segment the trunk accurately. The result of U-Net segmentation is better than that of SegNet and PSPNet, but there is still inaccurate trunk segmentation effect. DeepLabV3+ has outperformed the previous ones from all aspects of segmentation, but there is mis-segmentation of trunk gaps. SEMD model solves this problem very well.

#### 4.1.2. Experiment on Standing Trees with Complex Background

The comparison of standing tree segmentation under complex backgrounds using different methods are shown in Figure 10. In the complex background standing tree image segmentation experiment, due to the presence of natural environment noise, such as grass, soil, or incomplete green vegetation in the background, the segmentation becomes more difficult. 

Compared with the other five segmentation methods, SEMD reduces the calculation time imports an attention mechanism to make full use of the feature dimension, reduces the acquisition of erroneous features, and improves the detail processing at the edge of tree trunks. The overall standing tree segmentation effect is significantly enhanced, which is conducive to subsequent factor calculation, reduces manual intervention, and solves problems, such as low operation efficiency and inaccurate segmentation.

### 4.2. Comparision of Models

#### 4.2.1. Experiment on Standing Trees with Simple Background

According to the evaluation indicators described in Section 3.3.2, the SEMD model is compared with the other five algorithms, FCN, SegNet, U-Net, PSPNet, and the original DeepLabV3+, on the standing tree dataset, and the results are shown in Table 5.

Compared with the five other algorithmsunder the simple background condition, the MIoU of SEMD is greater than that of other methods no matter the image includes a single plant or multiple plants, which is improved to 93.22% and 89.21%, the MPA is increased to 96.71% and 94.23%, for the cases of a single tree and multiple trees, respectively.

#### 4.2.2. Experiment on Standing Trees with Complex Background

Compared with the cases of simple backgrounds, the performances of standing tree segmentation under complex backgrounds are lower, but the SEMD model still achieves better segmentation results in both single plant and multi-plant standing trees, as is shown in Table 6. Using SEMD model, MIoU is increased to 87.45% and 86.36%, and the MPA is increased to 94.72% and 92.23%, for the cases of a single tree and multiple trees, respectively. 

The PSNR values of three randomly selected samples with different standing tree backgrounds under different models are shown in Table 7.

As is shown in Table 7, SEMD has better performance in the segmentation results, which is almost 5dB higher than the other models on average, which proves that the model has high feasibility in the segmentation of standing trees.

### 4.3. Ablation Experiments

Ablation experiments represent one of the key factors in evaluating model quality. In this study, ablation experiments were performed using the same standing tree dataset. The experimental results are shown in Table 8, in which SE-M-D means the original DeepLabV3 without MobileNet and SENet, and SE-M+D means the version that Xception is replaced by MobileNet without SENet. It can be seen from the table that SEMD model not only reduces the computational complexity of the network and shortens the training running time, but also improves the segmentation accuracy.

SEMD model improves the useful information on the feature level and suppresses the generation of useless features for the standing tree segmentation task, so that SEMD model can be better optimized and the branch has better segmentation performance, which provides strong algorithm support for the subsequent automation of stumbling factor estimation.

### 4.4. Computational Complexity

In this section, we compare the proposed SEMD model with the remaining five models at training and inference time. Table 9 presents the training and inference times based on measurements on the hardware infrastructure described in Section 3.1. 

Because these methods were trained with the same optimizer and learning rates, these results were highly dependent on the depth of the network and the size of the chosen number. For example, SegNet is relatively fast in both training and inference, but its results in evaluation metrics are relatively low. Because of its simple network structure and fewer parameters, U-Net can also achieve faster training time, but the effect is slightly worse than SEMD. The original DeepLabv3+ is deeper than other networks.Therefore, it takes longer for training and inference than other networks. The SEMD network is based on the improvement of DeepLabV3+. Although its depth is deep, its parameters are few, so the network is one of the fastest networks in the training phase.

### 4.5. Robustness Test

To test the robustness of SEMD model, using experiments with different input image sizes, the performances of SEMD model are examined with different input sizes and the same other parameter settings to Section 3.3.3, and the results are shown in Table 10. 

It can be seen from the table that the SEMD model has similar performances under different input sizes, which indicates the robustness of SEMD model.

## 5. Conclusions

Focus on the semantic segmentation of standing tree images with complex backgrounds, a lightweight network segmentation model based on deep learning, SEMD, is proposed in this paper. It integrates the DeepLabV3+ model with the optimized lightweight network MobileNet, and an attention mechanism module SENet, so as to expand the receptive field, suppress the acquisition of erroneous feature information, shorten the training time, and improve the segmentation accuracy and robustness. Based on the dataset constructed from the standing tree images, the SEMD standing tree segmentation model was tested and compared with other methods. In simple background and complex background, SEMD improved the MIoU of single standing tree to 93.22% and 89.21%, and MIoU of multiple standing trees to 87.45% and 86.36%, respectively, with a loss function of 0.13. 

Although SEMD proposed in this paper can achieve a higher segmentation effect of multiple standing trees in complex backgrounds, the segmentation of standing trees in dark areas is inaccurate in the case of insufficient light, which will be studied and solved in our future works.

## Figures and Tables

**Figure 1 sensors-22-06663-f001:**
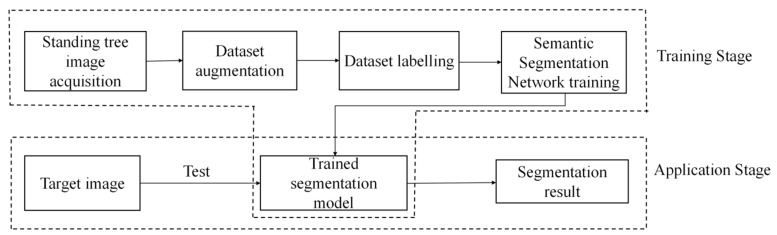
Flowchart of standing tree image segmentation.

**Figure 2 sensors-22-06663-f002:**
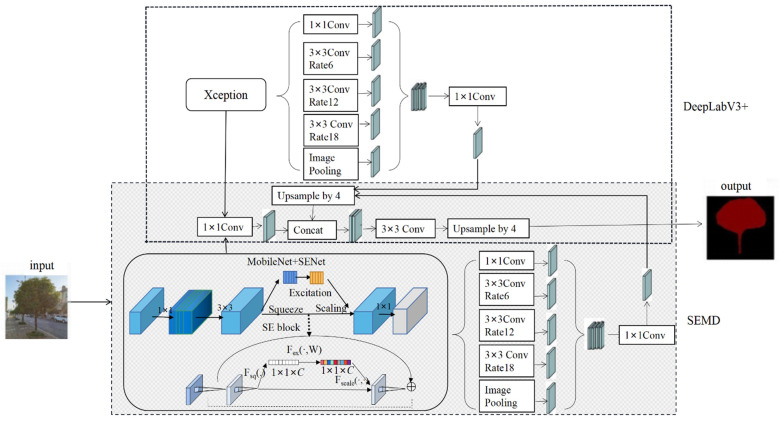
Overall architecture of the SEMD model.

**Figure 3 sensors-22-06663-f003:**
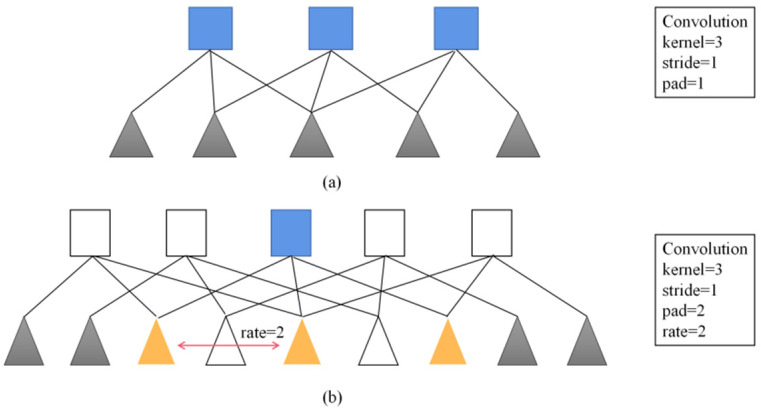
One-dimensional view of atrous convolution. (**a**) Convolution. (**b**) Atrous convolution.

**Figure 4 sensors-22-06663-f004:**
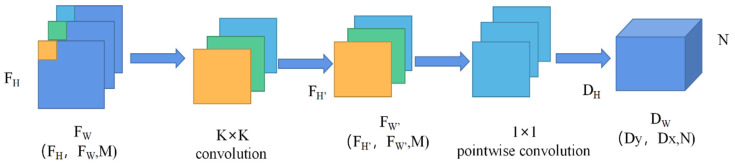
Depthwise separable convolution.

**Figure 5 sensors-22-06663-f005:**
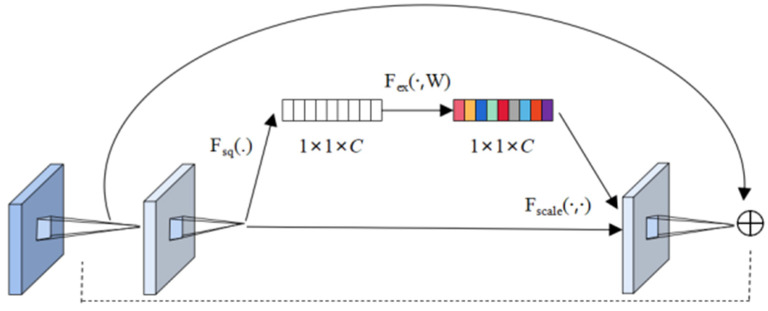
Structure of SENet.

**Figure 6 sensors-22-06663-f006:**
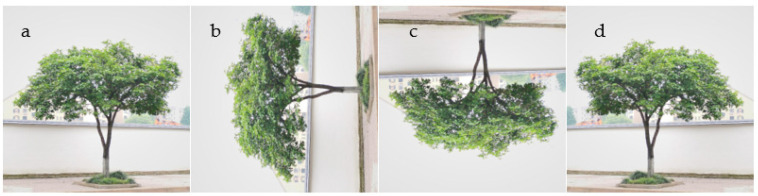
Data augmentation. (**a**) Original image. (**b**) Rotated image. (**c**) Flipped vertically. (**d**) Flipped horizontally.

**Figure 7 sensors-22-06663-f007:**
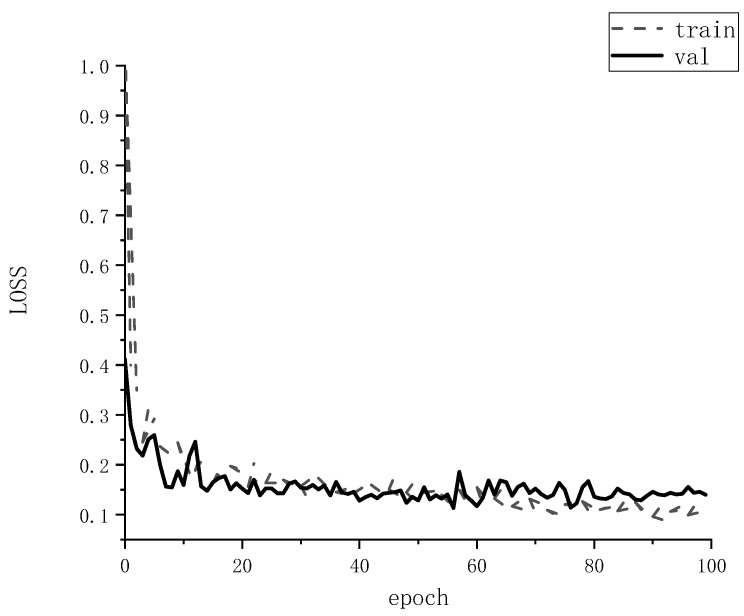
Loss function diagram of standing tree segmentation.

**Figure 8 sensors-22-06663-f008:**
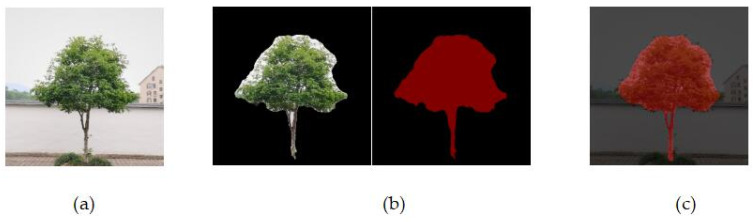
Overlay of segmentation results. (**a**) Original image. (**b**) Segmentation process. (**c**) Mask overlay.

**Figure 9 sensors-22-06663-f009:**
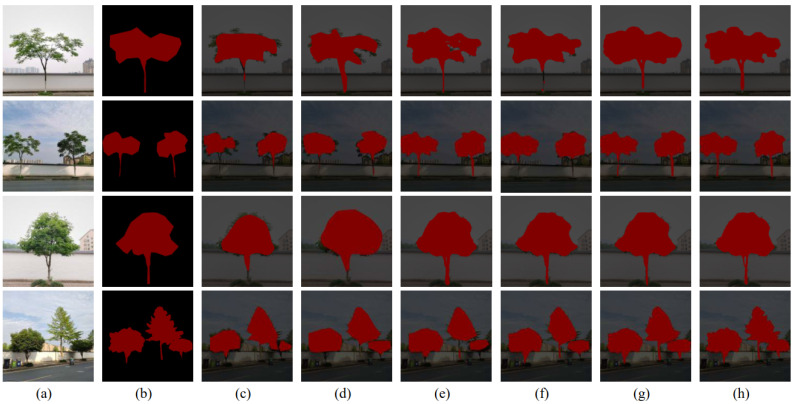
Comparison of standing tree segmentation under simple backgrounds. (**a**) Original image. (**b**) Ground-truth. (**c**) FCN. (**d**) SegNet. (**e**) U-Net. (**f**) PSPNet. (**g**) DeepLabV3. (**h**) SEMD.

**Figure 10 sensors-22-06663-f010:**
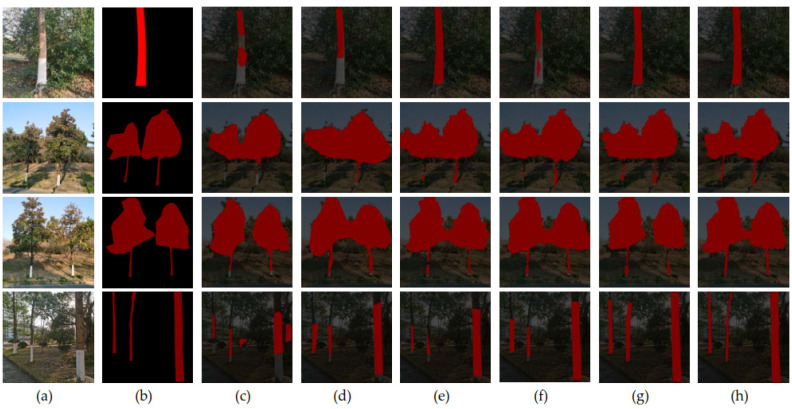
Comparison result of standing tree segmentation under complex backgrounds. (**a**) Original image. (**b**) Ground-truth. (**c**) FCN. (**d**) SegNet. (**e**) U-Net. (**f**) PSPNet. (**g**) DeepLabV3. (**h**) SEMD.

**Table 1 sensors-22-06663-t001:** MobileNetV2 network structure.

Input	Operator	t	c	n	s
224^2^ × 3	Conv2d	-	32	1	2
112^2^ × 32	Bottleneck	1	16	1	1
112^2^ × 16	Bottleneck	6	24	2	2
56^2^ × 24	Bottleneck	6	32	3	2
28^2^ × 32	Bottleneck	6	64	4	2
14^2^ × 64	Bottleneck	6	96	3	1
14^2^ × 96	Bottleneck	6	160	3	2
7^2^ × 160	Bottleneck	6	320	1	1
7^2^ × 320	Conv2d 1 × 1	-	1280	1	1
7^2^ × 1280	Avgpool 7 × 7	-	-	1	-
1 × 1 × 1280	Conv2d 1 × 1	-	k	-	

**Table 2 sensors-22-06663-t002:** MobileNetV2+ SENet network structure.

Input	Operator	t	c	n	s
224^2^ × 3	Conv2d	-	32	1	2
112^2^ × 32	Bottleneck	1	16	1	1
112^2^ × 16	Bottleneck	6	24	2	2
56^2^ × 24	Bottleneck	6	32	3	2
28^2^ × 32	Bottleneck	6	64	4	2
14^2^ × 64	Bottleneck	6	96	3	1
14^2^ × 96	Bottleneck	6	160	3	2
7^2^ × 160	Bottleneck	6	320	1	1

**Table 3 sensors-22-06663-t003:** Experimental software and hardware configuration.

Project	Detail
CPU	AMD Ryzen 7 5800H with Radeon Graphics @3.20 GHz
RAM	16 GB
Operating system	Windows 11 64-bit
CUDA	CUDA 11.3
Data processing	Python 3.6

**Table 4 sensors-22-06663-t004:** The hyper-parameters.

Project	Value
Epoch	100
Batch size	8
Lr	5 × 10^−5^
Input-shape	512 × 512

**Table 5 sensors-22-06663-t005:** Performance comparison in a simple background.

Model	Category	MIoU (%)	MPA (%)
FCN	Single tree	73.11	91.23
	Multiple trees	71.52	90.46
SegNet	Single tree	75.36	91.78
	Multiple trees	74.89	90.08
U-Net	Single tree	87.60	93.82
	Multiple trees	86.43	92.13
PSPNet	Single tree	73.23	80.36
	Multiple trees	66.35	85.73
DeepLabV3+	Single tree	92.15	95.57
	Multiple trees	85.62	94.85
SEMD	Single tree	93.22	96.71
	Multiple trees	89.21	94.23

**Table 6 sensors-22-06663-t006:** Performance comparison under complex background.

Model	Category	MIoU (%)	MPA (%)
FCN	Single tree	72.64	86.75
	Multiple trees	71.35	85.32
SegNet	Single tree	74.83	87.01
	Multiple trees	73.69	86.47
U-Net	Single tree	76.48	85.79
	Multiple trees	81.46	89.22
PSPNet	Single tree	72.69	82.34
	Multiple trees	70.62	81.78
DeepLabV3+	Single tree	83.18	90.17
	Multiple trees	79.23	90.46
SEMD	Single tree	87.45	94.72
	Multiple trees	86.36	92.23

**Table 7 sensors-22-06663-t007:** PSNR (dB) comparison of different models.

Background	Sample	FCN	SegNet	U-Net	PSPNet	DeepLabV3+	SEMD
Simple	1	15.76	16.75	17.49	16.49	20.59	23.42
	2	15.32	15.86	17.06	15.95	20.94	23.96
	3	16.77	15.34	19.98	16.86	21.38	25.42
Complex	1	14.69	15.98	16.23	14.76	20.46	26.32
	2	14.32	15.33	15.46	14.23	19.58	22.20
	3	15.13	14.62	15.74	14.06	19.63	22.48

**Table 8 sensors-22-06663-t008:** Comparison results of different model architectures.

Model	MIoU (%)	Speed (s)	Size (MB)
SE-M-D	85.21	0.53	94.30
SE-M+D	88.63	0.27	22.30
SEMD	90.35	0.16	22.00

**Table 9 sensors-22-06663-t009:** Average processing time for each method.

Model	Training Time (h:min)	Inference Time (s)
FCN	1:25	0.55
SegNet	1:06	0.63
U-Net	0:53	0.23
PSPNet	2:16	0.65
DeepLabV3+	1:32	0.53
SEMD	1:03	0.16

**Table 10 sensors-22-06663-t010:** Comparison results of different input-shape.

Input Shape	MIoU (%)	Speed (s)
512 × 512	90.35	0.16
321 × 321	89.96	0.12
415 × 415	90.07	0.15

## Data Availability

Not applicable.

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
