# Peer review of "Automatic Segmentation of Standing Trees from Forest Images Based on Deep Learning"

_sensors, 2022, doi:10.3390/s22176663_

Round 1
Reviewer 1 Report
The paper is titled – “Automatic Segmentation of Standing Trees from Forest Images Based on Deep Learning”. The paper proposes a method of image segmentation based on the DeepLabV3+ framework with SENet and MobileNet. The paper makes some contributions and is recommended for publication after minor revisions.
The reviewer has the following comments for the authors:
1. The paper reads well in most places. However, the reviewer still recommends proofreading the paper to improve readability.
2. Remove the numbering in the abstract.
3. Some acronyms not defined example DBH lines 28, CRF line 108, BN line 116, etc
4. Ideas mentioned in lines 40 – 41 would benefit from the addition of references.
5. Texts in Figures 1 and 2 should be clear and Figure 1 larger. The caption for Figure 2 should be on the same page as the Figure.
6. Line 69 missing ‘with’. It should read “…carries on with…”
7. “The” in caps, line 118. Why?
8. I do not understand the relevance of “Till now….” as used in line 148. I would suggest rewriting the sentence.
9. Some intext variables in lines 200 – 205 appear as superscripts. Reformat them to look as they appeared in eq 2 – 4.
10. Large font lines 305 -306 should be reformatted, check the font sizes and style for eq 7, 8, and 9.
11. Was CV used? It is unclear if the values in tables 5 and 6 are average results from all steps or not. If the former, then also measurement errors should be indicated.
Author Response
请参阅附件。

Reviewer 2 Report
This paper handles automatic segmentation problem of standing trees from forest images based on deep learning. The topic sounds interesting, and some comments are as follows:
1. There is a problem with the capitalization of the text, and some professional terms are wrongly written.
2. The description of some theories uses too much text. Formulas and equations should be added appropriately, such as adding some related formulas when describing the principle of MobileNetV2.
3. The amount of pictures in this paper is not enough, leading to the reduction of readability.
4. Formula is part of the paper, please adjust the position of the formula and add suitable punctuation at the end.
5. DeepLabV3+ model has no limit on the size of the input data, but the size of all experimental images in this paper is still unified. Please change the size of the input images for experiments to further verify the robustness of your model.
6. The amount of trees in the forest images used in the experiment is not enough, it is recommended to add another series of images containing three or more trees.
7. The models used in the comparison experiment is insufficient. Please add more models for comparison.
8. Please add more research contents to discussion section.
9. Other image segmentation methods should also be discussed in the introduction section, e.g., active contour model, An active contour model driven by adaptive local pre-fitting energy function based on Jeffreys divergence for image segmentation, Expert Systems with Applications; A hybrid active contour model based on pre-fitting energy and adaptive functions for fast image segmentation, Pattern Recognition Letters
Round 2
Reviewer 2 Report
The authors have addressed all my comments.